# Downregulation of LAMB3 Altered the Carcinogenic Properties of Human Papillomavirus 16-Positive Cervical Cancer Cells

**DOI:** 10.3390/ijms25052535

**Published:** 2024-02-22

**Authors:** Warattaya Wattanathavorn, Masahide Seki, Yutaka Suzuki, Supranee Buranapraditkun, Nakarin Kitkumthorn, Thanayod Sasivimolrattana, Parvapan Bhattarakosol, Arkom Chaiwongkot

**Affiliations:** 1Medical Microbiology Interdisciplinary Program, Graduate School, Chulalongkorn University, Bangkok 10330, Thailand; wwattanathavorn@gmail.com (W.W.); bhparvapan@gmail.com (P.B.); 2Center of Excellence in Applied Medical Virology, Faculty of Medicine, Chulalongkorn University, Bangkok 10330, Thailand; 3Department of Computational Biology and Medical Sciences, Graduate School of Frontier Sciences, The University of Tokyo, Kashiwa 277-8561, Chiba, Japan; mseki@edu.k.u-tokyo.ac.jp (M.S.); ysuzuki@edu.k.u-tokyo.ac.jp (Y.S.); 4King Chulalongkorn Memorial Hospital, Bangkok 10330, Thailand; bsuprane2001@yahoo.com; 5Division of Allergy and Clinical Immunology, Department of Medicine, Faculty of Medicine, 1873 Rama IV Road, Chulalongkorn University, Bangkok 10330, Thailand; 6Center of Excellence in Vaccine Research and Development (Chula Vaccine Research Center—Chula VRC), Faculty of Medicine, Chulalongkorn University, Bangkok 10330, Thailand; 7Department of Oral Biology, Faculty of Dentistry, Mahidol University, Bangkok 10400, Thailand; nakarinkit@gmail.com; 8Department of Microbiology, Faculty of Public Health, Mahidol University, Bangkok 10400, Thailand; thanayod.sasi@gmail.com; 9Department of Microbiology, Faculty of Medicine, Chulalongkorn University, 1873 Rama IV Road, Pathumwan, Bangkok 10330, Thailand

**Keywords:** transcriptomic, gene ontology, cervical cancer, LAMB3

## Abstract

Nearly all cervical cancer cases are caused by infection with high-risk human papillomavirus (HR-HPV) types. The mechanism of cervical cell transformation is related to the powerful action of viral oncoproteins and cellular gene alterations. Transcriptomic data from cervical cancer and normal cervical cells were utilized to identify upregulated genes and their associated pathways. The laminin subunit beta-3 (LAMB3) mRNAwas overexpressed in cervical cancer and was chosen for functional analysis. The LAMB3 was predominantly expressed in the extracellular region and the plasma membrane, which play a role in protein binding and cell adhesion molecule binding, leading to cell migration and tissue development. LAMB3 was found to be implicated in the pathway in cancer and the PI3K-AKT signaling pathway. LAMB3 knockdown decreased cell migration, invasion, anchorage-dependent and anchorage-independent cell growth and increased the number of apoptotic cells. These effects were linked to a decrease in protein levels involved in the PI3K-AKT signaling pathway and an increase in p53 protein. This study demonstrated that LAMB3 could promote cervical cancer cell migration, invasion and survival.

## 1. Introduction

According to the GLOBOCAN 2020 database, cervical cancer is a type of tumor that is more common in developing nations. Cervical cancer is the fourth most frequent cancer in women worldwide, with 604,127 new cases diagnosed and more than half dying [1]. It was reported that high-risk HPV types (HR-HPVs) were detected in approximately 99.7% of cervical cancer cases [2,3]. HPV16 was detected in up to 50% of cervical cancer cases [4,5,6,7]. The deregulated expression of HPV E6 and E7 oncoproteins is linked to tumor progression. E7 protein can bind with retinoblastoma (pRB), resulting in the release and activation of the transcription factors (E2F) and driving keratinocytes into S-phase, while E6 protein can degrade the tumor suppressor (p53), preventing apoptosis [8,9,10,11]. Consequently, HPV oncoprotein overexpression promotes genome instability, the accumulation of mitotic defects and cellular gene alterations in infected cells, all of which contribute to cell transformation and tumor progression. HR-HPV infections can also result in dysregulated cellular gene expressions and the activation of cancer-associated pathways mediated by viral oncoproteins, which drive the malignant transformation of cervical tissue [12,13,14,15].

Previous studies revealed cellular gene alterations in cervical cancer, such as pathways involved in several cancer pathways, genes involved in cell proliferation, viral carcinogenesis and pathways in cancer [16,17,18]. Altogether, they indicated that the upregulated genes in cervical cancer samples were involved in some critical pathways that are necessary for cancer cell survival. It has been known that HR-HPV oncoproteins induced cellular gene alterations that are involved in cervical carcinogenesis [19]. Some cellular genes were upregulated in only HPV16-positive cervical cancer cells but not in HPV-negative cervical cancer cells [20,21].

The role of laminin subunit beta-3 (LAMB3) has been studied extensively in several cancers, for example, pancreatic cancer [22], head and neck squamous cell carcinoma [23], thyroid cancer [24], and colorectal cancer [25]. Laminin-332, an extracellular matrix protein, interacts with cell-surface receptors and cytoplasmic signaling pathways to regulate a variety of cellular functions [26]. One of the subunits in laminin-332 is LAMB3 [27]. Several studies reported that LAMB3 overexpression is linked to cancer hallmarks, including cell proliferation, migration, invasion, and cancer progression via the PI3K/Akt signaling pathway, AKT-FOXO3/4, and c-MET/Akt signals [22,24,28]. In addition, it has also been found that HPV L1 protein can bind with a component of the extracellular matrix (ECM), such as laminin-332 (laminin 5) [29,30,31]. Association between HPV virions and laminin-332 in the ECM could transmit HPV virions to receptors on adjacent cells. This mechanism may also refer to laminin-332 as an “extracellular trans-receptor” [32]. One study reported that LAMB3 was highly expressed in HPV16-positive cervical cancer cell lines (SiHa), and laboratory investigations revealed that HPV16E6 protein induced the expression of LAMB3, while miR-218 expression in the SiHa cell line inhibited LAMB3 protein expression [33]. The involvement of LAMB3 in proliferation, migration, invasion and other cancer properties requires additional investigation. The current study aimed to investigate the functional role of LAMB3 in cervical cancer progression using HPV16-positive cervical cancer cell lines (CaSki and SiHa).

## 2. Results

### 2.1. Analysis of Differentially Expressed Genes (DEGs)

Cellular gene expression analysis comparing normal cervical samples and cervical cancer samples was performed using two datasets. Genes with log2 fold change of more than 2 were selected, and a total of 779 and 582 upregulated genes were identified in GEPIA CESC and GSE223804 datasets, respectively (Appendix A). There were 71 overlapping DEGs that upregulated in both datasets (Appendix A). The significant KEGG pathways included metabolic pathways, pathways in cancer, human papillomavirus infection and the PI3K-Akt signaling pathway (Appendix A). The *LAMB3* gene was selected for further functional analysis due to its involvement in pathways in cancer and the PI3K-Akt signaling pathway. Gene ontology analysis of the LAMB3 revealed that it was mostly located in the cell surface, plasma membrane and extracellular region, and it played a role in protein binding, integrin binding and signal receptor binding, leading to cell adhesion, cell migration and tissue development (Appendix A).

Real-time RT-PCR results showed that the LAMB3 mRNA had significantly higher levels of gene expression in the HPV16-positive cervical cancer cell lines (SiHa and CaSki) when compared to HEK293 and C33A cell lines, as shown in Appendix A. Our results are consistent with previous results [21,33]; therefore, CaSki and SiHa cell lines were further used for functional LAMB3 investigation.

### 2.2. The Optimal Concentration of siRNA for LAMB3 Silencing in Cervical Cancer Cell Lines

An siRNA concentration that could suppress LAMB3 mRNA expression by more than 70% in both the CaSki and SiHa cell lines was selected for further experiments. The siRNA concentration of 20 pmole/well of a 24 well-plate with transfection for 72 h was appropriate for silencing of LAMB3 mRNAexpression in both the SiHa and CaSki cervical cancer cell lines (Figure 1a,b).

### 2.3. Effect of LAMB3 siRNA Knockdown on Cervical Cancer Cell Proliferation

The results show that cell proliferation was slightly decreased in the LAMB3 siRNA knockdown SiHa cell line when compared to NTC (Figure 2a), whereas no significant difference was observed in the CaSki cell line (Figure 2b). This indicates that LAMB3 may have little effect or may not be a significant factor in cell proliferation in the SiHa and CaSki cell lines.

### 2.4. Effect of LAMB3 siRNA Knockdown on Cervical Cancer Cell Migration and Invasion

A wound-healing assay revealed that 24 h after scratching, the cell migration of LAMB3 siRNA knockdown SiHa cells was significantly slower than NTC, as observed by the open wound (Figure 3a,b). Meanwhile, when compared to NTC, there was a decrease in the cell migratory ability of LAMB3 siRNA-knockdown CaSki cells observed at 24 h post-transfection; the difference could not be observed at 48–72 h (Figure 3c,d). Cell migration was also investigated by the Boyden chamber assay; it was significantly decreased in LAMB3 siRNA knockdown cells when compared to NTC (Figure 4a,c). The invasion capacity of both cervical cancer cells was significantly decreased in LAMB3 siRNA knockdown when compared to NTC (Figure 4b). Thus, knockdown of LAMB3 inhibited cell migration and invasion in SiHa and CaSki cells.

### 2.5. Effect of LAMB3 siRNA Knockdown on the Anchorage-Dependent Growth of Cervical Cancer Cell Lines

We investigated the influence of LAMB3 on the colony formation capacity of SiHa and CaSki cervical cancer cell lines. There was a slight decrease in LAMB3 siRNA knockdown SiHa cells when compared to NTC (Figure 4e,f). However, the results revealed that the colony-forming efficiency of CaSki cells decreased significantly after LAMB3 siRNA knockdown (Figure 4g,h).

### 2.6. LAMB3 siRNA Knockdown Reduced the Anchorage-Independent Growth of HPV16-Positive Cervical Cancer Cell Lines

The effect of LAMB3 siRNA knockdown on anchorage-independent cell growth was investigated using the soft agar assay. The colony formation in soft agar is shown in Figure 5a. The number and size of colonies were significantly reduced in LAMB3 siRNA knockdown CaSki and SiHa cells when compared to the NTC (Figure 5b and Figure 5c, respectively).

### 2.7. Effect of LAMB3 siRNA Knockdown on Cell Apoptosis and Cell Cycle Progression

Our findings demonstrate that LAMB3 siRNA knockdown SiHa and CaSki cells significantly decreased the G1 phase and particularly increased the subG1 phase when compared to the NTC (Figure 6b,c, Appendix A). LAMB3 siRNA knockdown SiHa and CaSki cells also resulted in significantly increased apoptotic cells (Figure 7b,c, Appendix A). This elevated SubG1 population was associated with an increase in apoptotic cells. This indicated that LAMB3 knockdown had an effect on cervical cancer cell survival.

### 2.8. Effect of LAMB3 siRNA Knockdown on Expression of Proteins Involved in PI3K-AKT Signaling Pathway and p53 Protein

One of the major signaling pathways involved in the control of cell proliferation, migration and survival is the PI3K-AKT signaling pathway. We investigated the molecular mechanisms underlying the loss of cell fitness resulting from LAMB3 siRNA knockdown. As shown in Figure 8, AKT, pAKT, PI3K and pPI3K were decreased in LAMB3 siRNA knockdown SiHa and CaSki cells when compared to the NTC. In contrast, protein levels of p53 were increased in LAMB3 siRNA knockdown cells when compared to NTC. Cervical cancer cell lines with LAMB3 siRNA knockdown showed decreased hallmarks of cancer properties, such as cell proliferation, migration and invasion, that might be associated with downregulation of proteins involved in the PI3K/AKT signaling pathway. The upregulation of the p53 protein was associated with increased cell apoptosis of LAMB3 siRNA knockdown cell lines.

## 3. Discussion

In the present study, differential gene expression analysis was performed to identify cellular genes involved in cervical carcinogenesis. KEGG pathway analysis showed that the upregulated LAMB3 in cervical cancer samples is involved in many significant pathways, such as pathways in cancer and the PI3K-AKT signaling pathway. It was also associated with a crucial biological pathway that promotes the survival, migration and adhesion of cervical cancer cells. Altogether, the transcriptome profiles of altered cellular genes in cervical cancer cells shown in the present study were similar to those in previous research, demonstrating the validity of the selected data [16,17,18,19,21]. However, bioinformatic analysis is only a tool to predict the function and correlation of groups of genes. Functional analysis of upregulated genes is required to prove their important role in cervical carcinogenesis.

The cellular genes that play an important role in tumor initiation are mainly classified into at least three groups, including proto-oncogenes, tumor suppressor genes and genes involved in DNA repair mechanisms [34]. Several studies have reported on the role of upregulated cellular genes in cervical carcinogenesis; for example, one study identified novel critical genes, including AURKB, KYNU and LCP1, that were implicated in the HPV-induced carcinogenesis of several HPV-related cancers [35], SALL4 promotes tumorigenicity of cervical cancer cells [36], Nurr1 promotes cell proliferation, migration, invasion and anchorage-independent cell growth [37], and TREX1 and HR-HPV16 E7 oncoprotein collaborate in the cellular pathway to effectively enhance cervical carcinogenesis and progression [38]. According to one study, the protein sialyltransferase I (ST6Gal-I) is involved in tumor metastasis as well as cisplatin resistance in cervical cancer. Silencing the ST6Gal-I gene in the cervical cancer cell lines increased susceptibility to cisplatin, promoted cell apoptosis and inhibited invasion [39]. It was reported that HPV16E6 protein induced the expression of LAMB3 [33].

The functional results could indicate that LAMB3 upregulation in both cervical cancer cell lines (SiHa and CaSki) plays a major role in promoting cervical cancer cell invasion and migration but has little effect on cervical cancer cell proliferation. Another study reported that silencing of some genes in cervical cancer cell lines (SiHa and CaSki) had an effect only on cell migration and invasion but not on cell viability, such as the Ezrin gene [40]. For lung cancer cell lines (A549), FRAS1 knockdown reduced cell migration and invasion but not cell proliferation [41]. The present study demonstrated that colony formation was significantly decreased after LAMB3 siRNA knockdown in both cervical cancer cell lines. This indicates that LAMB3 upregulation enhanced clonogenic potential and self-renewal ability and might be involved in anoikis resistance properties in cervical cancer cell lines. It was noticeable that the SiHa cell line showed stronger anchorage-independent growth characteristics than the CaSki cell line, while both cell lines displayed excellent anchorage-dependent growth characteristics in the present study.

The PI3K-AKT signaling cascade promotes cell survival, proliferation, migration and energy metabolism, which has a significant impact on many essential cellular biology characteristics [42]. The present study showed that LAMB3 was upregulated in cervical cancer samples and was involved in the PI3K-AKT signaling pathway, which was previously reported to have carcinogenesis involvement in other cancer cells [22,24,28]. Protein–protein interaction analysis using the STRING database revealed interactions between the LAMB3 protein and other cellular proteins that are grouped in pathways related to cell migration and matrix degradation processes that promote cell invasion. LAMB3 was found to interact with other proteins, including CD44, ITGA6 and ITGB4, which interact with PI3K and AKT proteins (Figure 9). It was observed that CD44 triggered the MAPK/ERK and PI3K/AKT signaling pathways [43]. The PI3K/AKT signaling pathway was reported to be involved in the epithelial–mesenchymal transition (EMT) and matrix metalloproteinase 9 (MMP9) activity, promoting the invasiveness and metastasis of cancer cells [44,45]. It is proposed that siRNA knockdown of LAMB3 affects the cellular phenotypic features mentioned above by downregulating the protein partners involved in the PI3K-AKT signaling pathway. Increasing apoptosis in LAMB3 siRNA knockdown cancer cells might be due to an increased level of p53 protein in the present study, which is consistent with a previous study [46].

In summary, overexpression of proteins in extracellular matrix (ECM) components and alteration of the normal ECM environment leads to tumor progression in various cancers. LAMA3, LAMB3 and LAMC2 genes encode for laminin-332, which is related to tumor invasiveness and metastasis and is considered a factor of poor prognosis. LAMB3 has been shown to be elevated in various cancers, including cervical cancer. One study of colorectal cancer reported that LAMB3 overexpression was activated by BRD2/acetylated ELK4 complex; furthermore, LAMB3 overexpression activated AKT and resulted in degradation of FOXO3/4, a tumor suppressor protein [25]. *LAMB3* gene is considered a pro-metastatic gene in various cancers such as lung cancer, gastric cancer, pancreatic cancer, colorectal cancer and papillary thyroid cancer [22,24,25,47]. Laminin-332 promotes tumor progression by interacting with cell surface receptors, such as integrins, and affecting the related downstream signaling pathways that are involved in cancer cell migration and invasion, such as laminin-332-α3β1 integrin signaling (which can activate the FAK/Src/Rac1 pathway), laminin-332-α6β4 integrin signaling (which can activate the Ras/Raf/MEK/Erk pathway), or the PI3K/AKT pathway by recruiting receptor tyrosine kinases (RTKs) [25,48]. LAMB3 upregulation can also activate AKT (protein kinase B) activity via PI3K protein [22,49]. Phosphorylated AKT contributes to the phosphorylation of proteins located in the plasma membrane, the nucleus or the cytoplasm, leading to activation of cellular pathways such as the PI3K/AKT/mTOR signaling pathway and the PI3K/AKT/MAPK signaling pathway [50]. The molecular forms of laminin-332 were associated with cell anchoring, stable adherence or cell movement. The proposed mechanism related to LAMB3 involved in cancer cell migration and invasion is that laminin-332 overexpression in cancer cells can be switched from an anchorage protein to a free soluble protein favoring migration and invasion of cancer; laminin β3 chain and collagen VII interaction promotes carcinogenesis via activation of PI3K signaling pathway, which results in inhibition of apoptosis and increased tumor invasion. Laminin-332 can also bind to integrins α6β4 or α3β1, leading to tumor migration. To maintain cancer cell survival, laminin-binding β1 integrins can also activate the GTPases Rac1 and Cdc42; as a consequence, this causes rearrangement of the cytoskeleton, leading to cell migration [51]. One study showed that LAMB3 upregulation was due to promoter hypomethylation [47].

## 4. Materials and Methods

### 4.1. Cervical Cancer Cell Lines

HPV16-positive cervical cancer cell lines (SiHa (HTB-35) and CaSki (CRL-1550)), HPV-negative cervical cancer cell line (C33A, HTB-31) and human embryonic kidney 293T cell line (HEK, CRL-3216) were purchased from the American Type Culture Collection (ATCC, Manassas, VA, USA). They were maintained in DMEM high glucose medium (cat no. SH30022.02, Hyclone) supplemented with 10% fetal bovine serum (Gibco, Waltham, MA, USA) and 100 U/mL penicillin and streptomycin (Cytiva, Lane Cove West, Australia). All cells were cultured at 37 °C and 5% CO_2_. The present study was approved by the Institutional Review Board of the Faculty of Medicine, Chulalongkorn University (COE No. 005/2022, IRB No. 027/65), and the Institutional Biosafety Committee (IBC) of the Faculty of Medicine, Chulalongkorn University (MDCU-IBC026/2021).

### 4.2. Cellular Gene Expression Analysis

Differentially expressed transcripts comparing cervical cancer samples and normal cervical samples were identified. Two datasets (GSE223804 and cervical squamous cell carcinoma and endocervical adenocarcinoma (CESC)) were retrieved from GENBANK (http://www.ncbi.nlm.nih.gov/geo (accessed on 30 April 2023)) and the Gene Expression Profiling Interactive Analysis (GEPIA) database (http://gepia.cancer-pku.cn/index.html (accessed on 1 May 2023)). The overlapped significant Differentially Expressed Genes (DEGs) with log2 (fold change) > 2 (normal vs. cervical cancer samples) in both GSE223804 and GEPIA CESC were used for Kyoto Encyclopedia of Genes and Genomes (KEGG) pathway analysis (https://www.genome.jp/kegg/mapper/search.html (accessed on 1 May 2023)) to identify significant pathways. The selected gene was used for Gene Ontology (GO) analysis, including biological processes (BP), molecular pathways (MP) and cellular components (CC) based on STRING database (https://string-db.org/ (accessed on 1 May 2023)).

### 4.3. Real-Time Reverse Transcription Polymerase Chain Reaction (Real-Time RT-PCR)

The primer sequences were as follows: LAMB3 forward; 5’-AGCTTTCAGGCGATCTGGAG-3, LAMB3 reverse; 5′-GTCTCAGGCTTGGTCAGTCC-3′, GAPDH forward; 5-GCACCGTCAAGGCTGAGAAC-3 and GAPDH reverse; 5′-ATGGTGGTGAAGACGCCAGT-3′ were used to detect gene expression in RNA extracted from SiHa, CaSki, C33A and HEK cell lines. RNA extraction was performed using the RNeasy^®^ Mini Kit (QIAGEN, Valencia, CA, USA) following the manufacturer’s instructions. The 300 ng of extracted RNA was converted to cDNA using Superscript III Reverse Transcriptase (Invitrogen, Carlsbad, CA, USA). Real-time RT-PCR was performed using SSO Advanced Universal SYBR Green Supermix (Bio-Rad, Hercules, CA, USA). The gene expression difference was calculated by the delta-delta Ct method, and GAPDH was used as the housekeeping gene.

### 4.4. Optimization of siRNA Cell Transfection Conditions

The siRNA transfection was performed using Lipofectamine RNAi MAX reagent (Invitrogen, Carlsbad, CA, USA). LAMB3 siRNA sequences were as follows: sense: 5′-GUG UGU GCA AGG AGC AUG U(dTdT)-3′ and antisense: 5′-ACA UGC UCC UUG CAC ACA C(dTdT)-3′ synthesized by Dharmacon [23]. The negative control (NTC siRNA) was ON-TARGETplus Non-targeting Control siRNAs (catalog ID: D-001810-01-20, horizon). CaSki and SiHa cell lines were seeded at a density of 1 × 10^5^ cells/well in a 24-well plate supplemented with 10% FBS DMEM medium and incubated at 37 °C with 5% CO_2_ overnight. siRNAs were prepared at concentrations of 2.5, 5, 10, and 20 pmol/well and mixed with lipofectamine. The complex was added into each well and incubated for 72 h. Cells transfected with non-target control (NTC) or non-targeting siRNAs (D-001810-01-20, horizon) were used as a negative control at each concentration. Real-time RT-PCR and Western blot analysis were done to evaluate the expression levels of LAMB3 RNA and protein, respectively. An experiment to determine the time for optimal transfection was also performed, and incubation time points were 24, 48, 72 and 96 h.

### 4.5. Cell Proliferation Analysis by MTS Assay

The cell viability was determined using the MTS test. CaSki and SiHa cell lines were seeded at 1 × 10^5^ cells/well in a 24-well plate and incubated at 37 °C with 5% CO_2_ overnight before the experiment; cells were treated with 20 pmol siRNA. At 48 h post-transfection, transfected cells were seeded in a 96-well plate and incubated at 37 °C with 5% CO_2_ for 0, 24, 48, 72 and 96 h. The cell proliferation reagent MTS (CellTiter96 proliferation assay, Promega, Madison, WI, USA) was used, and the absorbance was measured at a wavelength of 490 nm using a microplate reader (Perkin Eimer, Waltham, MA, USA). The relative cell proliferation was calculated. Cells transfected with NTC siRNA served as a negative control.

### 4.6. Wound-Healing Assay

LAMB3 knockdown CaSki and SiHa cell lines were seeded at 2 × 10^5^ cells/well in a 24-well plate and incubated at 37 °C with 5% CO_2_ overnight. Next, the cell-free gap in the monolayer cells was created by scratching with a sterilized 200 µL micropipette tip, and the culture medium was changed in each well. The wound area was measured at 0, 24, 48 and 72 h using Pixit Pro.A4 program. The percentage of cell migration through the gap (% wound healing) was calculated.

### 4.7. Cell Migration and Cell Invasion Assays by Boyden Chamber Assay

For the cell migration assay, at 48 h post-transfection, 1 × 10^5^ transfected cells were resuspended in 200 µL serum-free DMEM medium and placed in the upper chamber with polycarbonate membrane pore size 8.0 µm (REF353097, Falcon, Monfalcone, Italy) placed in a 24-well plate (SKU: 353047, Falcon), with the lower chamber containing 700 µL of 10% FBS DMEM medium as a chemoattractant, and incubated at 37 °C with 5% CO_2_ overnight. Non-migratory cells on top of the membrane were removed using cotton-tipped applicators. Then, migratory cells were fixed with 70% ethanol for 10 min, stained with 0.2% crystal violet solution for 10 min and washed gently. Migratory cells were counted under an inverted microscope.

The cell invasion assay was performed by using the transwell chamber coated with 35 µL of ECM at a concentration of 250 µg/mL and incubated at 37 °C with 5% CO_2_ for 2 h. Then, 1 × 10^5^ transfected cells were resuspended in 200 µL serum-free DMEM medium and placed in the chamber coated with ECM gel placed in a 24-well plate (SKU: 353047, Falcon). Next, 700 µL of 10% FBS DMEM medium as a chemoattractant was added in the lower chamber and incubated at 37 °C with 5% CO_2_ overnight. Non-invasive cells on top of the membrane were removed using cotton-tipped applicators. The invasive cells were fixed with 70% ethanol for 10 min, stained with 0.2% crystal violet solution for 10 min and washed gently. Invasive cells were counted under an inverted microscope. The percentages of migration and invasion were calculated; cells transfected with non-targeting siRNAs (NTC) served as a negative control.

### 4.8. Colony Formation Assay

The anchorage-dependent cell growth ability of transfected cells was performed. For SiHa cell lines, at 48 h post-transfection, transfected cells were seeded at 500 cells in 3 mL of medium per well in 6-well plates. For CaSki cell lines, transfected cells were seeded at 1000 cells in 3 mL of medium per well in 6-well plates. The cells were then incubated for 14 days at 37 °C with 5% CO_2_. Colonies were stained with 0.005% crystal violet for 1 h, and colonies were counted.

### 4.9. Soft Agar Colony Formation Assay

An experiment to determine the anchorage-independent cell growth ability of transfected cells was performed. The bottom layer was prepared by mixing equal volumes (1:1 ratio) of 1% Noble agar and 2× DMEM medium supplemented with 10% FBS. Next, 1.5 mL of the mixture was added into each well of a 6-well plate and left for 30 min at room temperature. For the upper layer, 1 × 10^4^ transfected cells were suspended in 2 mL of 2× DMEM medium supplemented with 10% FBS, then mixed with 2 mL of 0.6% agarose and immediately plated on top of base agar. Then, 200 µL of 1× DMEM supplemented with 10% fetal bovine serum was added into each well and incubated at 37 °C with 5% CO_2_ for approximately 4 weeks with regular addition of culture medium every 2–3 days. Finally, colonies were counted under an inverted microscope.

### 4.10. Apoptosis and Cell Cycle Progression Analysis by Flow Cytometry

At 72 h post-transfection, 1 × 10^6^ transfected cells were washed 2 times with cold BioLegend cell staining buffer (420201, BioLegend, San Diego, CA, USA) and resuspended in 100 µL of Annexin V Binding buffer (42220, BioLegend). The whole suspension was then treated with 5 µL of APC Annexin V (640920, BioLegend) and 10 µL of Propidium Iodide solution (421301, BioLegend) and incubated in the dark for 15 min. After that, 400 µL of Annexin V Binding buffer (42220, BioLegend) was added, and the apoptotic cells were measured by BD FACSCalibur™ Flow Cytometer (BD Biosciences, San Jose, CA, USA).

For cell cycle progression, 1 × 10^6^ transfected cells were washed 1 time with cold PBS. Then, 1 mL of cold 70% ethanol was added dropwise to the cell pellet and fixed at 4 °C for 30 min. After that, the fixed cells were collected and resuspended in 1 mL of PBS. The fixed cells were then treated with 50 µL of 100 µg/mL RNase A (Thermo Scientific, Boston, MA, USA), 425 µL of cell staining buffer (420201, BioLegend), and 25 µL of Propidium Iodide solution (421301, BioLegend). Finally, cell cycle profile analysis was measured by using BD FACSCalibur™ Flow Cytometer (BD Biosciences).

### 4.11. Western Blot Analysis

The p53 protein and proteins involved in the PI3K/AKT signaling pathway, including AKT, p-AKT(473), P13K and p-PI3K, were analyzed by Western blot. Transfected SiHa and CaSki cervical cancer cell lines were used for protein extraction using cOmplete™ Lysis-M EDTA-free (Roche, Sigma Aldrich, St. Louis, MO, USA), and protein concentration was measured by using a nanodrop spectrophotometer (Eppendorf BioSpectrometer^®^, Macquarie Park, Australia, basic). The samples (200 µg) were loaded and separated using 10% SDS-polyacrylamide gel and transferred to a nitrocellulose membrane (BioRAD). The membranes were blocked with a 3% BSA solution. Each membrane was incubated with a specific antibody, including LAMB3 antibody (ab97765, 1:1000, Abcam, Cambridge, UK), PI3K antibody (p110 4255, 1:1000, cell signaling technology), pPI3K antibody (p-p85 4228, 1:1000, cell signaling technology), p53 antibody (p53 9282, 1:1000, cell signaling technology), and GAPDH antibody (sc-47724, 1:200, Santacruz Biotechnology). Secondary HRP-conjugated rabbit anti-mouse (ab205719) and HRP-conjugated anti-rabbit IgG (ab205718) antibodies (Abcam) were used. The protein was determined using a chemiluminescent detection method using the ChemiDoc XRS+ System (BIO-RAD). The intensity of the protein band was assessed using Image LabTM 6.0 software.

### 4.12. Statistical Analysis

All in vitro experiments were performed in either duplicate or triplicate on three independent experiments. The statistical significance was analyzed to investigate the differences between groups using the unpaired *t*-test. Data are presented as mean + standard deviation (SD), and *p*-value ≤ 0.05 was considered statistically significant. The data were analyzed using GraphPad Prism 9 (Dotmatics, San Diego, CA, USA).

## 5. Conclusions

This study is the first to demonstrate that LAMB3 downregulation affects the significant characteristics of HPV16-positive cervical cancer cells, including decreased migration, invasion and metastasis. Additionally, cervical cancer cell apoptosis was observed. These results might be related to downregulation in the proteins in the PI3K-AKT signaling pathway and an increased level of p53 protein. Based on these data, LAMB3 appears to be a potential therapeutic drug target and prognosis of cancer progression.

## Figures and Tables

**Figure 1 ijms-25-02535-f001:**
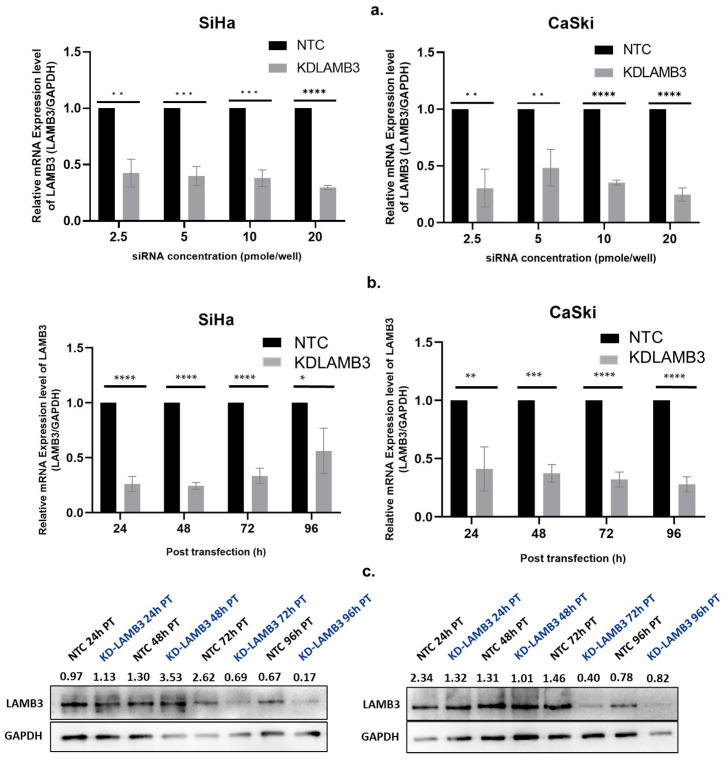
Optimization of siRNA concentration and time point for LAMB3 silencing in cervical cancer cell lines. (**a**) LAMB3 mRNA expression in SiHa and CaSki cell lines at different siRNA concentrations ( 2.5, 5, 10, and 20 pmol/well). (**b**,**c**) LAMB3 mRNA and protein expression in SiHa and CaSki cell lines at different time points examined by real-time RT-PCR and Western blot analysis, respectively. GAPDH was used as a housekeeping gene. An experiment was performed in triplicate on three independent experiments. NTC; non-target control (non-targeting siRNAs), KDLAMB3; LAMB3 siRNA knockdown. SD is shown as error bars. Ns, non-significant: *p* > 0.05, *: *p* ≤ 0.05, **: *p* ≤ 0.01, ***: *p* ≤ 0.001, ****: *p* ≤ 0.0001.

**Figure 2 ijms-25-02535-f002:**
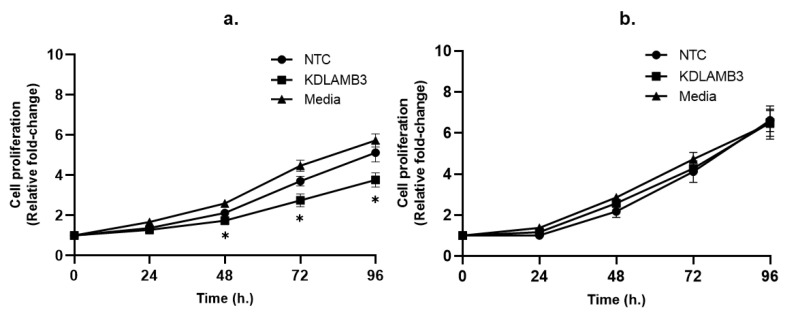
Effect of LAMB3 siRNA knockdown on proliferation of cervical cancer cell lines measured by MTS assay. Cell proliferation was measured at 0, 24, 48, 72 and 96 h. Relative fold change of cell proliferation of SiHa (**a**) and CaSki (**b**) cell lines with and without LAMB3 siRNA knockdown was calculated. An experiment was performed in triplicate on three independent experiments. NTC; non-target control (non-targeting siRNAs), KDLAMB3; LAMB3 siRNA knockdown. SD is shown as error bars. Ns, non-significant: *p* > 0.05, *: *p* ≤ 0.05.

**Figure 3 ijms-25-02535-f003:**
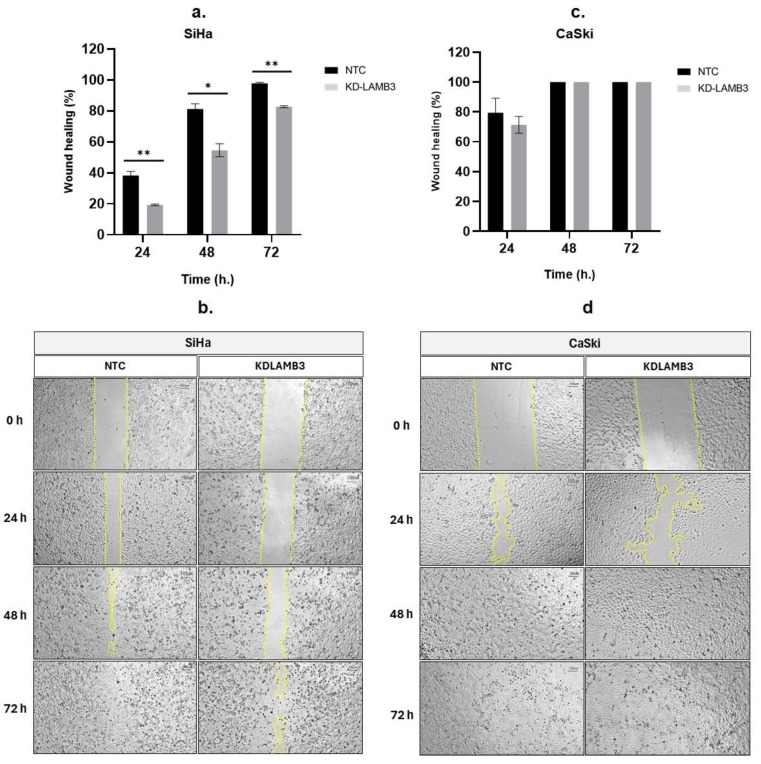
Effect of LAMB3 siRNA knockdown on cell migration ability of cervical cancer cell lines detected by the scratch wound-healing assay. Knockdown of LAMB3 in SiHa and CaSki cells inhibited cell migration. The percentage of wound healing of LAMB3 siRNA-transfected SiHa and CaSki cells was compared to non-targeting control, which was measured by surface area (**a**,**c**). The representative images of scratch wound healing, showing cell migration into the open area were shown in (**b**,**d**). An experiment was performed in triplicate on three independent experiments. NTC; non-target control (non-targeting siRNAs), KDLAMB3; LAMB3 siRNA knockdown. SD is shown as error bars. Ns, non-significant: *p* > 0.05, *: *p* ≤ 0.05, **: *p* ≤ 0.01.

**Figure 4 ijms-25-02535-f004:**
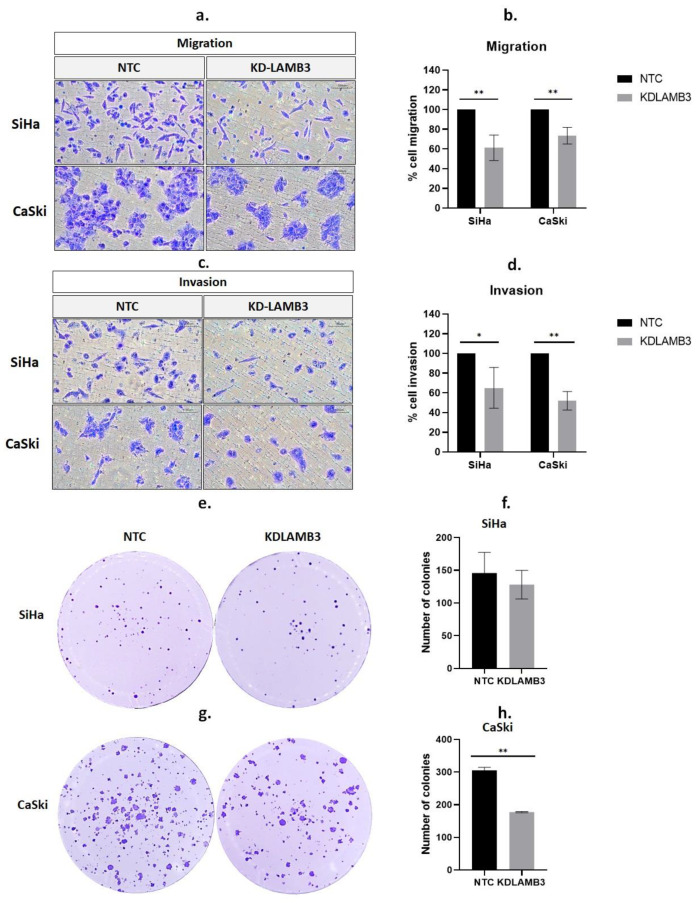
Effect of LAMB3 siRNA knockdown on cell migration, invasion and anchorage-dependent cell growth of cervical cancer cell lines. The percentage of migration (**a**,**b**) and invasion (**c**,**d**) in LAMB3 siRNA knockdown SiHa and CaSki cells was compared to NTC detected by Boyden chamber assay. The number of SiHa (**e**,**f**) and CaSki (**g**,**h**) colonies with LAMB3 siRNA knockdown was compared to NTC. An experiment was performed in triplicate on three independent experiments. NTC; non-target control (non-targeting siRNAs), KDLAMB3; LAMB3 siRNA knockdown. SD is shown as error bars. Ns, non-significant: *p* > 0.05, *: *p* ≤ 0.05, **: *p* ≤ 0.01.

**Figure 5 ijms-25-02535-f005:**
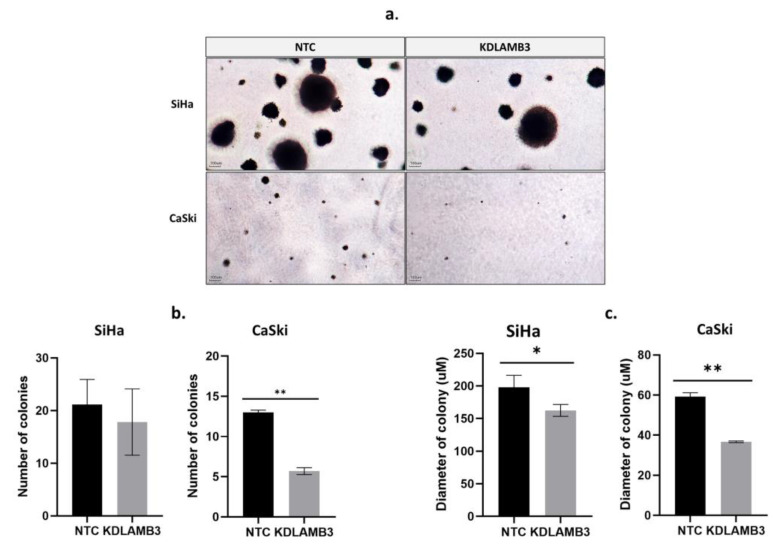
Effect of LAMB3 siRNA knockdown on the anchorage-independent cell growth of cervical cancer cell lines. In (**a**), colony formation in soft agar is demonstrated. The (**b**) number and (**c**) diameter of colonies of SiHa and CaSki cells with and without LAMB3 siRNA knockdown were measured. An experiment was performed in duplicate on three independent experiments. NTC; non-target control (non-targeting siRNAs), KDLAMB3; LAMB3 siRNA knockdown. SD is shown as error bars. Ns, non-significant: *p* > 0.05, *: *p* ≤ 0.05, **: *p* ≤ 0.01.

**Figure 6 ijms-25-02535-f006:**
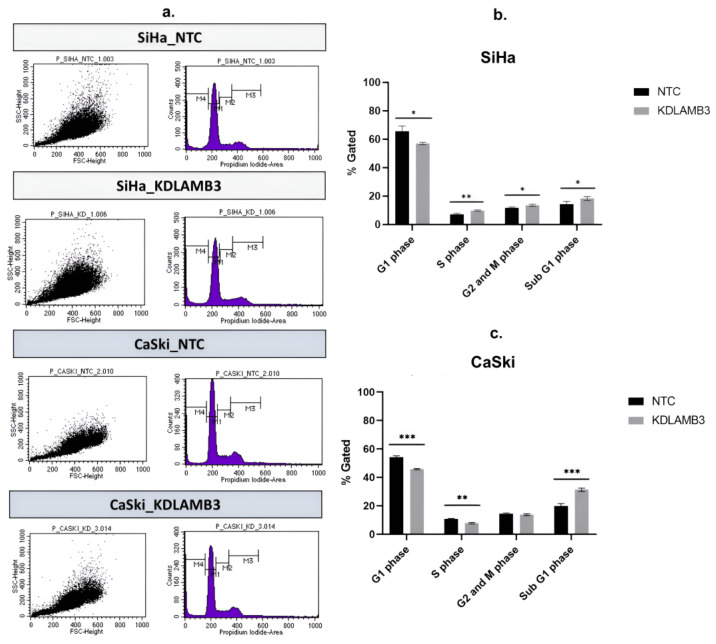
Effect of LAMB3 siRNA knockdown on the cell cycle profile in cervical cancer cell lines. The flow cytometry results at 72 h post-transfection (**a**). The percentage of cell cycle population in cell cycle profile of SiHa cells (**b**) and CaSki cells (**c**) with and without LAMB3 siRNA knockdown was compared to NTC siRNA control cells. An experiment was performed in triplicate on three independent experiments. NTC; non-target control (non-targeting siRNAs), KDLAMB3; LAMB3 siRNA knockdown. SD is shown as error bars. Ns, non-significant: *p* > 0.05, *: *p* ≤ 0.05, **: *p* ≤ 0.01, ***: *p* ≤ 0.001.

**Figure 7 ijms-25-02535-f007:**
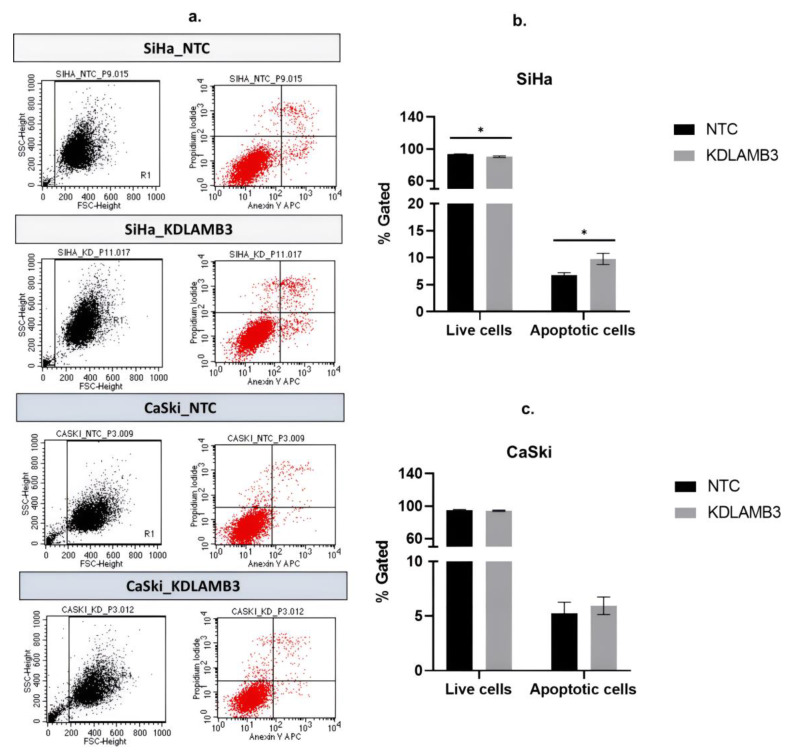
Effect of LAMB3 siRNA knockdown on the apoptosis in cervical cancer cell lines. The flow cytometry results at 72 h post-transfection (**a**). The percentage of apoptosis of SiHa cells (**b**) and CaSki cells (**c**) with LAMB3 siRNA knockdown was compared to NTC. An experiment was performed in triplicate on three independent experiments. NTC; non-target control (non-targeting siRNAs), KDLAMB3; LAMB3 siRNA knockdown. SD was shown as error bars. Ns, non-significant: *p* > 0.05, *: *p* ≤ 0.05.

**Figure 8 ijms-25-02535-f008:**
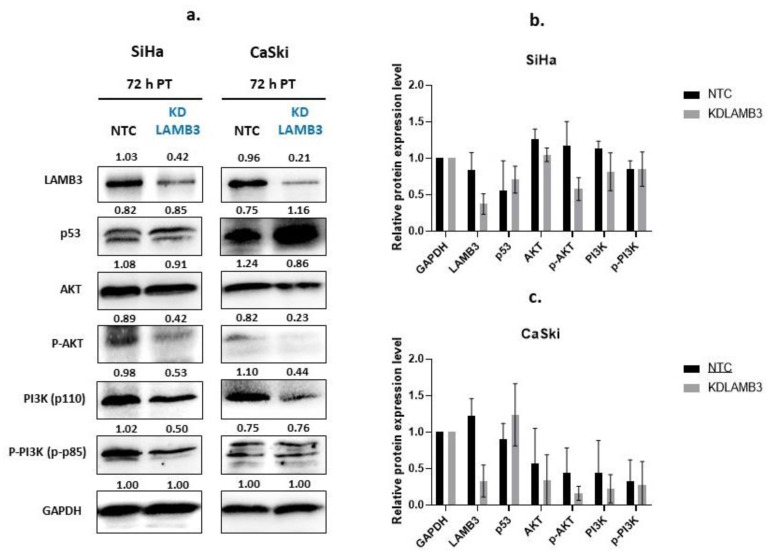
Effect of LAMB3 siRNA knockdown on expression levels of proteins involved in the PI3K-AKT signaling pathway and p53 in cervical cancer cell lines. The Western blot results at 72 h post-transfection (**a**). The relative protein expression levels of SiHa cells (**b**) and CaSki cells (**c**) with and without LAMB3 siRNA knockdown were calculated. An experiment was performed in duplicate on three independent experiments. NTC; non-target control (non-targeting siRNAs), KDLAMB3; LAMB3 siRNA knockdown. SD is shown as error bars.

**Figure 9 ijms-25-02535-f009:**
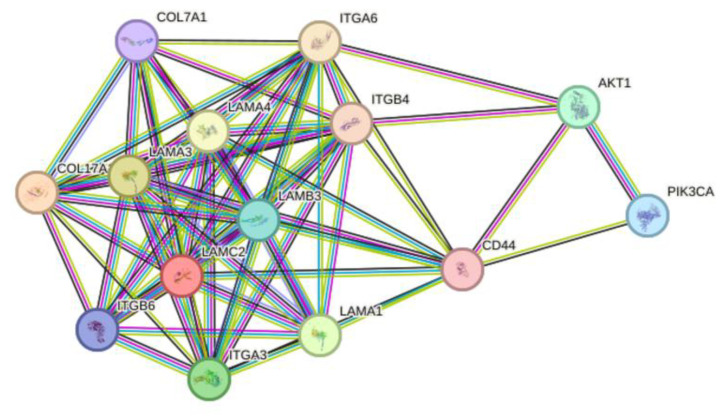
The evaluation of protein–protein interaction of LAMB3 by STRING database. The lines represent the different levels of evidence. Predicted interaction is represented by a green line (neighborhood evidence), a blue line (co-occurrence evidence) and a red line (gene fusion evidence). Known interaction is represented by a purple line (experimental evidence) and a light blue line (database evidence). The others are a yellow line (textmining evidence), a black line (coexpression evidence) and a gray line (protein homology).

## Data Availability

The datasets and materials used and/or analyzed during the current study are available from the corresponding author upon reasonable request.

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
