# Peer review of "Downregulation of LAMB3 Altered the Carcinogenic Properties of Human Papillomavirus 16-Positive Cervical Cancer Cells"

_ijms, 2024, doi:10.3390/ijms25052535_

Round 1

Reviewer 1 Report

Comments and Suggestions for Authors

Better understanding biology of many oncologic disorders seems to be a key factor of proper, targeted and personalized treatment. Due to high incidence of HPV infections and their huge impact on cancer development direction of the studies is justified. 

Revised paper is composed in clear manner. Methodology is correct and well-designed. References are in general ok. 

I have one remark. The authors stated that LAMB3 functional role in cervical cancer have never been reported, what is not true. There is study by Martinez at al. from 2008 published in Oncogene (PMID: 17998940). It is also not placed in ref. so I suppose the authors missed it. It should be introduced into references, into intro as well discussion... 

After all, this is well presented concise paper which after introduction my suggestion may be considered for publication in IJMS. 

Author Response

Dear Reviewer 

Thank you for your kind suggestion. In this revised version, we carefully rewrote this manuscript with more details as highlighted in the introduction and discussion.  We also added reference as recommended.  We have changed the sentence “LAMB3 functional role in cervical cancer have never been reported” to be “One study reported that LAMB3 was highly expressed in HPV16 positive cervical cancer cell lines (SiHa), and laboratory investigations revealed that HPV16E6 protein induced the expression of LAMB3, while miR-218 expression in SiHa cell line inhibited LAMB3 protein expression [33]. The involvement of LAMB3 in proliferation, migration, invasion and other cancer properties requires additional investigation. The current study aimed to investigate the functional role of LAMB3 in cervical cancer progression using HPV16 positive cervical cancer cell lines (CaSki and SiHa).”

Reviewer 2 Report

Comments and Suggestions for Authors

Manuscript Title: Downregulation of LAMB3 altered the carcinogenic properties of human papillomavirus 16 positive cervical cancer cells

Manuscript ID- IJMS-2873465

The manuscript is mainly focused on the cervical cancer which is one of the leading disease nowadays. The involvement of LAMB3 in promoting cervical cancer cell migration, invasion and survival has been discussed in this paper. This study is quite fit in the present context of cancer research however there are few mistakes needs to be rectified before moving further with decision.

Comment 1: It has been shown in the Real-time RT-PCR results that LAMB3 gene had significantly higher levels of gene expression in the HPV 16 positive cervical cancer cell line (SiHa and CaSki) as compared to HEK293 and C33A cell lines. It would be interesting to find the protein of LAMB3 gene acting on a particular target. The use of bioinformatics technique such as molecular docking is highly applicable to find the possible role of protein and knowing its mechanism at molecular level.

Comment 2: The LAMB3 gene can be translated and its protein can be obtained with three-dimensional structure to know its further mechanism.

Comment 3: “The LAMB3 down-regulation affects the significant characteristics of cervical cancer cells including decreased migration, invasion and metastasis” This statement can be well explained with supportive processes and information’s.

Comment 4: The conclusion part is too short. It must be rewritten to get exact idea of molecular activities happening during the cervical cancer.

Comment 5: Overall the manuscripts well written having potential data to be considered further but must me proofread to rectify grammatical errors before considering further. 

Comments on the Quality of English Language

Few grammatical mistakes needs to be corrected

Author Response

Reviewer 2

Manuscript Title: Downregulation of LAMB3 altered the carcinogenic properties of human papillomavirus 16 positive cervical cancer cells

Manuscript ID- IJMS-2873465

The manuscript is mainly focused on the cervical cancer which is one of the leading disease nowadays. The involvement of LAMB3 in promoting cervical cancer cell migration, invasion and survival has been discussed in this paper. This study is quite fit in the present context of cancer research however there are few mistakes needs to be rectified before moving further with decision.

Comment 1: It has been shown in the Real-time RT-PCR results that LAMB3 gene had significantly higher levels of gene expression in the HPV 16 positive cervical cancer cell line (SiHa and CaSki) as compared to HEK293 and C33A cell lines. It would be interesting to find the protein of LAMB3 gene acting on a particular target. The use of bioinformatics technique such as molecular docking is highly applicable to find the possible role of protein and knowing its mechanism at molecular level.

Our response: Thank you for your valuable suggestion. In this revised version, we have performed protein-protein interaction using STRING database, the result is shown in Figure 9 and is written in the discussion section, which is  “Protein-protein interaction analysis using the STRING database revealed in-teractions between the LAMB3 protein and other cellular proteins that are grouped in pathways related to cell migration and matrix degradation processes, which promote cell invasion. LAMB3 was found to interact with other proteins, including CD44, ITGA6 and ITGB4 which interact with PI3K and AKT proteins (Figure 9). It was observed that CD44 triggered the MAPK/ERK and PI3K/AKT signaling pathways [43].”

Comment 2: The LAMB3 gene can be translated and its protein can be obtained with three-dimensional structure to know its further mechanism.

Our response: Thank you for your valuable suggestion. In this revised version, we have performed protein-protein interaction using STRING database, the result is shown in Figure 9 and is written in the discussion section. For three-dimensional structure and molecular docking analysis will be our further work.

Comment 3: “The LAMB3 down-regulation affects the significant characteristics of cervical cancer cells including decreased migration, invasion and metastasis” This statement can be well explained with supportive processes and information’s.

Our response: Thank you for your valuable suggestion. In this revised version, we have added more information in conclusion part, which is “This study is the first to demonstrate that LAMB3 down-regulation affects the significant characteristics of HPV16 positive cervical cancer cells including decreased migration, invasion and metastasis. Additionally, cervical cancer cell apoptosis was also observed. These results might be related to downregulation in the proteins in the PI3K-AKT signaling pathway and an increased level of p53 protein Based on these data, LAMB3 appears to be a potential therapeutic drug target and prognosis of cancer progression”

Comment 4: The conclusion part is too short. It must be rewritten to get exact idea of molecular activities happening during the cervical cancer.

Our response: Thank you for your valuable suggestion. In this revised version, we have added more information in conclusion part, which is “This study is the first to demonstrate that LAMB3 down-regulation affects the significant characteristics of HPV16 positive cervical cancer cells including decreased migration, invasion and metastasis. Additionally, cervical cancer cell apoptosis was also observed. These results might be related to downregulation in the proteins in the PI3K-AKT signaling pathway and an increased level of p53 protein Based on these data, LAMB3 appears to be a potential therapeutic drug target and prognosis of cancer progression”

Comment 5: Overall the manuscripts well written having potential data to be considered further but must me proofread to rectify grammatical errors before considering further.

Our response: Thank you very much for suggestions, we have carefully proofread our manuscript.

Comments on the Quality of English Language

Few grammatical mistakes needs to be corrected

Our response: Thank you very much for suggestions, we have carefully proofread our manuscript.